# CORESETS FOR ACCELERATING
# INCREMENTAL GRADIENT METHODS

## ABSTRACT

Many machine learning problems reduce to the problem of minimizing an expected risk. Incremental gradient (IG) methods, such as stochastic gradient descent and its variants, have been successfully used to train the largest of machine learning models. IG methods, however, are in general slow to converge and sensitive to stepsize choices. Therefore, much work has focused on speeding them up by reducing the variance of the estimated gradient or choosing better stepsizes. An alternative strategy would be to select a carefully chosen subset of training data, train only on that subset, and hence speed up optimization. However, it remains an open question how to achieve this, both theoretically as well as practically, while not compromising on the quality of the final model. Here we develop CRAIG, a method for selecting a weighted subset (or coreset) of training data in order to speed up IG methods. We prove that by greedily selecting a subset $S$ of training data that minimizes the upper-bound on the estimation error of the full gradient, running IG on this subset will converge to the (near)optimal solution in the same number of epochs as running IG on the full data. But because at each epoch the gradients are computed only on the subset $S$, we obtain a speedup that is inversely proportional to the size of $S$. Our subset selection algorithm is fully general and can be applied to most IG methods. We further demonstrate practical effectiveness of our algorithm, CRAIG, through an extensive set of experiments on several applications, including logistic regression and deep neural networks. Experiments show that CRAIG, while achieving practically the same loss, speeds up IG methods by up to 10x for convex and 3x for non-convex (deep learning) problems.

## 1 INTRODUCTION

Mathematical optimization lies at the core of training large-scale machine learning systems, and is now widely used over massive data sets with great practical success, assuming sufficient data resources are available. Achieving this success, however, also requires large amounts of (often GPU) computing, as well as concomitant financial expenditures and energy usage (Strubell et al., 2019). Significantly decreasing these costs without decreasing the learnt system's resulting accuracy is one of the grand challenges of machine learning and artificial intelligence today (Asi & Duchi, 2019).

Training machine learning models often reduces to the problem of optimizing a regularized empirical risk function. Given a convex loss $l$, and a $\mu$-strongly convex regularizer $r$, one aims to find model parameter vector $x_*$ over the parameter space $\mathcal{X}$ that minimizes the loss $f$ over the training data $V$:

$$x_* \in \operatorname*{argmin}_{x \in \mathcal{X}} f(x), \quad f(x) := \sum_{i \in V} f_i(x) + r(x), \quad f_i(x) = l(x, (a_i, y_i)), \tag{1}$$

where $V = \{1, \ldots, n\}$ is an index set of the training data, and functions $f_i : \mathbb{R}^d \to \mathbb{R}$ are associated with training examples $(a_i, y_i)$, where $a_i \in \mathbb{R}^d$ is the feature vector, and $y_i$ is the point $i$'s label.

The standard Gradient Descent can find the minimizer of this problem, but requires repeated computations of the full gradient $\nabla f(x)$—sum of the gradients over all training data points/functions $i$—and is therefore prohibitive for massive data sets. This issue is further exacerbated in case of deep neural networks where gradient computations (backpropagation) are expensive. Incremental Gradient (IG) methods, such as Stochastic Gradient Descent and its accelerated variants, including SGD with momentum (Qian, 1999), Adagrad (Duchi et al., 2011), Adam (Kingma & Ba, 2014),

SAGA (Defazio et al., 2014), and SVRG (Johnson & Zhang, 2013) iteratively estimate the gradient on random subsets/batches of training data. While this provides an unbiased estimate of the full gradient, the randomized batches introduce variance in the gradient estimate (Hofmann et al., 2015), and therefore stochastic gradient methods are in general slow to converge (Johnson & Zhang, 2013; Defazio et al., 2014). The majority of the work speeding up IG methods has thus primarily focused on reducing the variance of the gradient estimate (SAGA (Defazio et al., 2014), SVRG (Johnson & Zhang, 2013), Katysha (Allen-Zhu, 2017)) or more carefully selecting the gradient stepsize (Adagrad (Duchi et al., 2011), Adadelta (Zeiler, 2012), Adam (Kingma & Ba, 2014)).

However, the direction that remains largely unexplored is how to carefully select a small subset $S \subseteq V$ of the full training data $V$, so that the model can only be trained on the subset $S$ while still (approximately) converging to the globally optimal solution (i.e., the model parameters that would be obtained if training/optimizing on the full $V$). If such a subset $S$ can be quickly found, then this would directly lead to a speedup of $|V|/|S|$ (which can be very large if $|S| \ll |V|$) per epoch of IG.

There are four main challenges in finding such a subset $S$. First is that a guiding principle to select $S$ is unclear. For example, selecting training points close to the decision boundary might allow the model to fine tune the decision boundary, while picking the most diverse set of data points would allow the model to get a better sense of the training data distribution. Second is that finding $S$ must be fast as otherwise identifying the set $S$ may take longer than the actual optimization, and so no overall speed-up would be achieved. Third is that finding a subset $S$ is not enough. One also has to decide on an ordering over $S$ and a gradient stepsize for each data point in $S$, as they affect the convergence. And last, while the method might work well empirically on some data sets, one also requires theoretical understanding and mathematical convergence guarantees.

Here we develop *CoResets for Accelerating Incremental Gradient descent (*CRAIG*)*, for selecting a subset of training data points to speed up training of large machine learning models. Our key idea is to directly approximate the gradient. That is, we aim to find a weighted and ordered subset $S$ of training data $V$ that is representative of the full gradient of $V$. We prove that the subset $S$ that minimizes an upper-bound on the error of estimating the full gradient of $V$ maximizes a submodular facility location function. As a result, the subset $S$ can be efficiently found using a fast greedy algorithm. A further benefit of our approach is that set $S$ is created incrementally which induces a natural ordering over data in $S$. Thus, rather than processing data points in a random or arbitrary order, CRAIG processes them using in the order provided by the procedure, which we show further speeds up the convergence of the method.

We also provide theoretical analysis of CRAIG and prove the convergence of our approach. In particular, for a $\mu$-strongly convex risk function and a subset $S$ selected by CRAIG that estimates the full gradient by an error of at most $\epsilon$, we prove that IG method with diminishing stepsize $\alpha_k = \alpha/k^\tau$ at epoch $k$ (with $0 < \tau < 1$ and $0 < \alpha$), converges to an $2R\epsilon/\mu$ neighborhood of the optimal solution at rate $\mathcal{O}(1/\sqrt{k})$. Here, $R = \min\{d_0, (r\gamma_{\max}C + \epsilon)/\mu\}$ where $d_0$ is the initial distance to the optimum, $C$ is an upper-bound on the norm of the gradients, $r = |S|$, and $\gamma_{\max}$ is the largest weight for the elements in the subset obtained by CRAIG. Moreover, we prove that if in addition to the strong convexity, component functions have smooth gradients, IG with the same diminishing stepsize on subset $S$ converges to a $2\epsilon/\mu$ neighborhood of the optimum solution at rate $\mathcal{O}(1/k^\tau)$.

The above convergence rates are the same as convergence rate of IG on $V$ for a strongly convex risk (and smooth component functions), and therefore IG on $S$ converges in the same number epochs as IG on the full $V$. But because every epoch only uses a subset $S$ of the data, it requires fewer gradient computations and thus leads to a $|V|/|S|$ speedup over traditional IG methods, while still (approximately) converging to the optimal solution. Furthermore, CRAIG only requires the knowledge of estimated gradient differences and does not involve any (exact) gradient calculations. Therefore, CRAIG can be used as a simple preprocessing step before IG starts and no additional storage or gradient calculations are required during IG, which makes CRAIG extremely practical. As such CRAIG can be used to speed up any existing IG methods, including IG, Adam, SAGA, SVRG as we show in the experiments section.

We demonstrate the effectiveness of CRAIG via an extensive set of experiments using logistic regression (a convex optimization problem) as well as training neural networks (non-convex optimization problems). We show that CRAIG speeds up incremental gradient methods, including SGD, SAGA, SVRG, Adam, Adagrad, and NAG. In particular, CRAIG while achieving practically the same loss

and accuracy as the underlying incremental gradient descent methods, speeds up gradient methods by up to 10x for convex and 3x for non-convex loss functions. We also demonstrate that the deliberate ordering scheme of the CRAIG algorithm significantly improves convergence time.

## 2 RELATED WORK

Convergence of IG methods has been long studied under various conditions (Zhi-Quan & Paul, 1994; Mangasariany & Solodovy, 1994; Bertsekas, 1996; Solodov, 1998; Tseng, 1998), however IG's convergence rate has been characterized only more recently (see (Bertsekas, 2015) for a survey). In particular, Nedić & Bertsekas (2001) provides a $\mathcal{O}(1/\sqrt{k})$ convergence rate for diminishing stepsizes $\alpha_k$ per epoch $k$ under a strong convexity assumption, and Gürbüzbalaban et al. (2015) proves a $\mathcal{O}(1/k^\tau)$ convergence rate with diminishing stepsizes $\alpha_k = \Theta(1/k^\tau)$ for $\tau \in (0,1]$ under an additional smoothness assumption for the components. While these works provide convergence on the full dataset, our analysis provides the same convergence rates on subsets obtained by CRAIG.

It has been empirically observed that ordering of data significantly affects the convergence rate of IG. However, finding a favorable ordering for IG has been a long standing open question. Among the few results are that of (Recht & Re, 2012) showing that without-replacement random sampling improves convergence of IG for least means squares problem, and the very recent result of (Gurbuzbalaban et al., 2017) showing that a Random Reshuffling (RR) method with iterate averaging and a diminishing stepsize $\Theta(1/k^\tau)$ for $\tau \in (1/2, 1)$ converges at rate $\Theta(1/k^{2\tau})$ with probability one in the suboptimality of the objective value, thus improving upon the $\Omega(1/k)$ rate of SGD. Contrary to the above randomized analysis, we propose the first deterministic ordering on the data points and empirically show that the ordering provided by CRAIG provides a significant speedup for the convergence of IG.

Techniques for speeding up SGD, are mostly focused on variance reduction techniques (Roux et al., 2012; Shalev-Shwartz & Zhang, 2013; Johnson & Zhang, 2013; Hofmann et al., 2015; Allen-Zhu et al., 2016), and accelerated gradient methods when the regularization parameter is small (Frostig et al., 2015; Lin et al., 2015; Xiao & Zhang, 2014). Very recently, Hofmann et al. (2015); Allen-Zhu et al. (2016) exploited neighborhood structure to further reduce the variance of stochastic gradient descent and improve its running time. Our CRAIG method and analysis are complementary to variance reduction and accelerated methods. CRAIG can be applied to these methods as well to speed them up (as we show in experiments).

## 3 CORESETS FOR INCREMENTAL GRADIENT DESCENT (CRAIG)

We proceed as follows: First, we define an objective function $L$ for selecting an optimal set $S$ of size $r$ that best approximates the gradient of the full training dataset $V$ of size $n$. Then, we show that $L$ can be turned into a submodular function $F$ and thus $S$ can be efficiently found using a greedy algorithm. Crucially, we also show that the approximation error between the estimated and the true gradient can be efficiently minimized in a way that is independent of the actual optimization procedure and thus CRAIG can simply be used as a preprocessing step before the actual optimization starts.

Incremental gradient methods aim at estimating the full gradient $\sum_{i \in V} \nabla f_i(x)$ over $V$ by iteratively making a step based on the gradient of every function $f_i$. Our key idea in CRAIG is that if we can find a small subset $S$ such that the weighted sum of the gradients of its elements closely approximates the full gradient over $V$, we can apply IG only to the set $S$ (with stepsizes equal to the weight of the elements in $S$), and we should still converge to the (approximately) optimal solution, but much faster.

Specifically, our goal in CRAIG is to find the smallest subset $S \subseteq V$ and corresponding per-element stepsizes $\gamma_j > 0$ that approximate the full gradient with an error at most $\epsilon > 0$ for all the possible values of the optimization parameters $x \in \mathcal{X}$.[1]

$$S^* = \underset{S \subseteq V, \gamma_j \geq 0 \ \forall j}{\operatorname{argmin}} |S|, \quad \text{s.t.} \quad \max_{x \in \mathcal{X}} \| \sum_{i \in V} \nabla f_i(x) - \sum_{j \in S} \gamma_j \nabla f_j(x) \| \leq \epsilon. \tag{2}$$

Given such an $S^*$ and associated weights $\{\gamma\}_j$, we are guaranteed that gradient updates on $S^*$ will be similar to the gradient updates on $V$ regardless of the value of $x$.

---

[1]Note that in the worst case we may need $|S^*| \approx |V|$ to approximate the gradient. However, as we show in experiments, in practice we find that a small subset is sufficient to accurately approximate the gradient.

Unfortunately, directly solving the above optimization problem is not feasible, due to two problems. Problem 1: Eq. (2) requires us to calculate the gradient of all the functions $f_i$ over the entire space $\mathcal{X}$, which is too expensive and would not lead to overall speedup. In other words, it would appear that solving for $S^*$ is as difficult as solving problem (1), as it involves calculating $\sum_{i \in V} \nabla f_i(x)$ for various $x \in \mathcal{X}$. And Problem 2: even if calculating the normed difference between the gradients in Eq. (2) would be fast, as we discuss later finding the optimal subset $S^*$ in NP-hard. In the following, we address the above two challenges and discuss how we can quickly find a near-optimal subset $S$.

### 3.1 UPPER-BOUND ON THE ESTIMATION ERROR

We first address Problem 1, i.e., how to quickly estimate the error/discrepancy of the weighted sum of gradients of functions $f_j$ associate with data points $j \in S$, vs the full gradient, for every $x \in \mathcal{X}$.

Let $S$ be a subset of $r$ data points. Furthermore, assume that there is a mapping $\varsigma_x : V \rightarrow S$ that for every $x \in \mathcal{X}$ assigns every data point $i \in V$ to one of the elements $j$ in $S$, i.e., $\varsigma_x(i) = j \in S$. Let $C_j = \{i \in [n] | \varsigma(i) = j\} \subseteq V$ be the set of data points that are assigned to $j \in S$, and $\gamma_j = |C_j|$ be the number of such data points. Hence, $\{C_j\}_{j=1}^r$ form a partition of $V$. Then, for any arbitrary (single) $x \in \mathcal{X}$ we can write

$$\sum_{i \in V} \nabla f_i(x) \quad = \quad \sum_{i \in V} \left( \nabla f_i(x) - \nabla f_{\varsigma_x(i)}(x) + \nabla f_{\varsigma(i)}(x) \right) \tag{3}$$

$$= \quad \sum_{i \in V} \left( \nabla f_i(x) - \nabla f_{\varsigma_x(i)}(x) \right) + \sum_{j \in S} \gamma_j \nabla f_j(x). \tag{4}$$

Subtracting and then taking the norm of the both sides, we get an upper bound on the error of estimating the full gradient with the weighted sum of the gradients of the functions $f_j$ for $j \in S$. I.e.,

$$\| \sum_{i \in V} \nabla f_i(x) - \sum_{j \in S} \gamma_j \nabla f_j(x) \| \leq \sum_{i \in V} \| \nabla f_i(x) - \nabla f_{\varsigma_x(i)}(x) \|, \tag{5}$$

where the inequality follows from the triangle inequality. The upper-bound in Eq. (5) is minimized when $\varsigma_x$ assigns every $i \in V$ to an element in $S$ with most gradient similarity at $x$, or minimum Euclidean distance between the gradient vectors at $x$. That is: $\varsigma_x(i) \in \operatorname{argmin}_{j \in S} \| \nabla f_i(x) - \nabla f_j(x) \|$. Hence,

$$\min_{S \subseteq V} \| \sum_{i \in V} \nabla f_i(x) - \sum_{j \in S} \gamma_j \nabla f_j(x) \| \leq \sum_{i \in V} \min_{j \in S} \| \nabla f_i(x) - \nabla f_j(x) \|. \tag{6}$$

The right hand side of Eq. (6) is minimized when $S$ is the set of $r$ *medoids* (exemplars) for all the components in the gradient space. So far, we considered upper-bounding the gradient estimation error at a particular $x \in \mathcal{X}$. To bound the estimation error for all $x \in \mathcal{X}$, we consider a worst-case approximation of the estimation error over the entire parameter space $\mathcal{X}$. Formally, we define a distance metric $d_{ij}$ between gradients of $f_i$ and $f_j$ as the maximum normed difference between $\nabla f_i(x)$ and $\nabla f_j(x)$ over all $x \in \mathcal{X}$:

$$d_{ij} \triangleq \max_{x \in \mathcal{X}} \| \nabla f_i(x) - \nabla f_j(x) \|. \tag{7}$$

Thus, by solving the following minimization problem, we obtain the smallest weighted subset $S^*$ that approximates the full gradient by an error of at most $\epsilon$ for all $x \in \mathcal{X}$:[1]

$$S^* = \operatorname*{argmin}_{S \subseteq V} |S|, \quad \text{such that} \quad L(S) \triangleq \sum_{i \in V} \min_{j \in S} d_{ij} \leq \epsilon. \tag{8}$$

Note that Eq. (8) requires that the gradient error is bounded over $\mathcal{X}$. However, we show (Appendix B) for several classes of convex problems, including linear regression, ridge regression, logistic regression, and regularized support vector machines (SVMs), the normed gradient difference between data points can be efficiently boundedly approximated by (Allen-Zhu et al., 2016; Hofmann et al., 2015):

$$\forall x, i, j \quad \| \nabla f_i(x) - \nabla f_j(x) \| \leq d_{ij} \leq \max_{x \in \mathcal{X}} \mathcal{O}(\|x\|) \cdot \|a_i - a_j\| = \text{const.} \|a_i - a_j\|. \tag{9}$$

Note when $\|x\|$ is bounded for all $x \in \mathcal{X}$, i.e., $\max_{x \in \mathcal{X}} \mathcal{O}(\|x\|) < \infty$, upper-bounds on the Euclidean distances between the gradients can be pre-computed. This is crucial, because it means that estimation error of the full gradient can be efficiently bounded independent of the actual optimization problem (i.e., point $x$). Thus, these upper-bounds can be computed only once as a pre-processing step before any training takes place, and then used to find the subset $S$ by solving the optimization problem (8). We address upper-bounding the normed difference between gradients for deep models in Section 3.4.

---

**Algorithm 1** CRAIG (CoResets for Accelerating Incremental Gradient descent)

---

**Input:** Set of component functions $f_i$ for $i \in V = \{1, \cdots, n\}$.
**Output:** Subset $S \subseteq V$ with corresponding per-element stepsizes $\{\gamma\}_{j \in S}$, and an ordering $\sigma$.
1:  $S_0 \leftarrow \emptyset, s_0 = 0, i = 0$.
2:  **while** $F(S) < L(\{s_0\}) - \epsilon$ **do**
3:      $j \in \text{argmax}_{e \in V \setminus S_{i-1}} F(e|S_{i-1})$
4:      $S_i = S_{i-1} \cup \{j\}$
5:      $\sigma_i = j$
6:      $i = i + 1$
7:  **end while**
8:  **for** $j = 1$ to $|S|$ **do**
9:      $\gamma_j = \sum_{i \in V} \mathbb{I}\big[j = \text{argmin}_{s \in S} \max_{x \in \mathcal{X}} \|\nabla f_i(x) - \nabla f_s(x)\|\big]$
10: **end for**

---

### 3.2 THE CRAIG ALGORITHM

Optimization problem (8) produces a subset $S$ of elements with their associated weights $\{\gamma\}_{j \in S}$ or per-element stepsizes that closely approximates the full gradient. Here, we show how to efficiently approximately solve the above optimization problem in order to find a near-optimal subset $S$.

The optimization problem (8) is NP-hard as it involves calculating the value of $L(S)$ for all the $2^{|V|}$ subsets $S \subseteq V$. We show, however, to transform it into a *submodular set cover problem*, for which efficient approximation algorithms exist.

Formally, $F$ is submodular if $F(S \cup \{e\}) - f(S) \geq F(T \cup \{e\}) - F(T)$, for any $S \subseteq T \subseteq V$ and $e \in V \setminus T$. We denote the *marginal* utility of an element $s$ w.r.t. a subset $S$ as $F(e|S) = F(S \cup \{e\}) - F(S)$. Function $F$ is called *monotone* if $F(e|S) \geq 0$ for any $e \in V \setminus S$ and $S \subseteq V$. The submodular cover problem is defined as finding the smallest set $S$ that achieves utility $\rho$. Precisely,

$$S^* = \underset{S \subseteq V}{\text{argmin}} |S|, \quad \text{such that} \quad F(S) \geq \rho. \tag{10}$$

Although finding $S^*$ is NP-hard since it captures such well-known NP-hard problems as Minimum Vertex Cover, for many classes of submodular functions (Nemhauser et al., 1978; Wolsey, 1982), a simple greedy algorithm is known to be very effective. The greedy algorithm starts with the empty set $S_0 = \emptyset$, and at each iteration $i$, it chooses an element $e \in V$ that maximizes $\triangle(e|S_{i-1})$, i.e., $S_i = S_{i-1} \cup \{\text{argmax}_{e \in V} F(e|S_{i-1}) - F(S_{i-1})\}$. Greedy gives us a logarithmic approximation, i.e. $|S| \leq \big(1 + \ln(\max_e F(e|\emptyset))\big)|S^*|$. The computational complexity of the greedy algorithm is $\mathcal{O}(|V| \cdot |S|)$. However, its running time can be reduced to $\mathcal{O}(|V|)$ using stochastic algorithms (Mirzasoleiman et al., 2015a) and further improved using lazy evaluation (Minoux, 1978), and distributed implementations (Mirzasoleiman et al., 2015b; 2016).

Given a subset $S \subseteq V$, the facility location function quantifies the coverage of the whole data set $V$ by the subset $S$ by summing the similarities between every $i \in V$ and its closest element $j \in S$. Formally, facility location is defined as $F_{fl}(S) = \sum_{i \in V} \max_{j \in S} s_{i,j}$, where $s_{i,j}$ is the similarity between $i, j \in V$. The facility location function has been used in a number of applications, including scene and documents summarization (Simon et al., 2007; Lin & Bilmes, 2012).

By introducing an auxiliary element $s_0$ we can turn $L(S)$ in Eq. (8) into a monotone submodular facility location function,

$$F(S) = L(\{s_0\}) - L(S \cup \{s_0\}), \tag{11}$$

where $L(\{s_0\})$ is a constant. In words, $F$ measures the decrease in the estimation error associated with the set $S$ versus the estimation error associated with just the auxiliary element. It is easy to see that for suitable choice of $s_0$, maximizing $F$ is equivalent to minimizing $L$. Therefore, we apply the greedy algorithm to approximately solve the following problem to get the subset $S$ defined in 8:

$$S^* = \underset{S \subseteq V}{\text{argmin}} |S|, \quad \text{such that} \quad F(S) \geq L(\{s_0\}) - \epsilon. \tag{12}$$

At every step, the greedy algorithm selects an element that reduces the upper bound on the estimation error the most. In fact, the size of the smallest subset $S$ that estimates the full gradient by an error of at most $\epsilon$ depends on the structural properties of the data. Intuitively, as long as the marginal gains

of facility location are considerably large, we need more elements to improve our estimation of the full gradient. Having found $S$, the weight $\gamma_j$ of every element $j \in S$ is the number of components that are closest to it in the gradient space, and are used as stepsize of element $j \in S$ during IG. The pseudocode for CRAIG subset selection method is outlined in Algorithm 1.

### 3.3 Ordering on the elements of the subset

Notice that CRAIG creates subset $S$ incrementally one element at a time, which produces a natural order $\sigma$ to the elements in $S$. Adding the element with largest marginal gain $j \in \operatorname{argmax}_{e \in V} F(e|S_{i-1})$ improves our estimation from the full gradient by an amount bounded by the marginal gain. Formally, at every step $i$, we have $F(S_i) \geq (1 - e^{-i/|S|})F(S^*)$, and hence

$$\|\sum_{i \in V} \nabla f_i(x) - \sum_{j \in S} \gamma_j \nabla f_j(x)\| \leq \mathrm{cnt} - (1 - e^{-i/|S|})L(S^*). \tag{13}$$

Intuitively, the first elements of the ordering contribute the most to provide a close approximation of the full gradient and the rest of the elements further refine the approximation. Hence, the first incremental gradient updates gets us close to $x_*$, and the rest of the updates further refine the solution. We show experimentally that processing data points in the order of $S$ leads to faster convergence than when we consider elements in $S$ in a random order. We defer the formal proof to future work.

### 3.4 Application of CRAIG to Deep Networks

Incremental gradient methods, including SGD with momentum (Qian, 1999), Adam (Kingma & Ba, 2014) and Adagrad (Duchi et al., 2011) are widely used to train deep networks. As discussed, CRAIG selects a subset that closely approximates the full gradient, and hence can be also applied for speeding up training deep networks. The challenge here is that we cannot use inequality (9) to bound the normed difference between gradients for all $x \in \mathcal{X}$ and find the subset as a preprocessing step.

However, it has been shown that for neural networks, the variation of the gradient norms is mostly captured by the gradient of the loss w.r.t. the last layer (see Section 3.2 of (Katharopoulos & Fleuret, 2018)) that is often not expensive or only slightly more expensive than calculating the loss. In many cases, where we have cross entropy loss with soft-max as the last layer, the gradient of the loss w.r.t. the $i$-th input to the soft-max is simply $p_i - y_i$, where $p_i$ are logits (dimension $p - 1$ for $p$ classes) and $y$ is the one-hot encoded label. In this case, CRAIG does not need any backward pass or extra storage. That is, CRAIG can be applied at the beginning of every epoch to find a subset for that epoch. Note that, although CRAIG needs an additional $\mathcal{O}(|V| \cdot |S|)$ complexity (or $\mathcal{O}(|V|)$ using stochastic greedy) to find the subset $S$ at the beginning of every epoch, this complexity does not involve any (exact) gradient calculations and is negligible compared to the cost of backpropagations performed during the epoch. Hence, as we show in the experiments CRAIG is practical and salable.

## 4 Convergence Rate Analysis of CRAIG

The idea of CRAIG is to selects a subset that closely approximates the full gradient, and hence can be applied to speed up most IG variants as we show in our experiments. Here, we briefly introduce the original IG method, and then prove the convergence rate of IG applied to subsets found by CRAIG.

### 4.1 Incremental Gradient Methods (IG)

Incremental gradient (IG) methods are core algorithms for solving Problem (1) and are widely used and studied. IG aims at approximating the standard gradient method by sequentially stepping along the gradient of the component functions $f_i$ in a cyclic order. Starting from an initial point $x_0^1 \in \mathbb{R}^d$, it makes $k$ passes over all the $n$ components. At every epoch $k \geq 1$, it iteratively updates $x_{i-1}^k$ based on the gradient of $f_i$ for $i = 1, \cdots, n$ using stepsize $\alpha_k > 0$. Formally,

$$x_i^k = x_{i-1}^k - \alpha_k \nabla f_i(x_{i-1}^k), \qquad i = 1, 2, \cdots, n, \tag{14}$$

with the convention that $x_0^{k+1} = x_n^k$. Note that for a closed and convex subset $\mathcal{X}$ of $\mathbb{R}^d$, the results can be projected onto $\mathcal{X}$, and the update rule becomes

$$x_i^k = P_{\mathcal{X}}(x_{i-1}^k - \alpha_k \nabla f_i(x_{i-1}^k)), \qquad i = 1, 2, \cdots, n, \tag{15}$$

where $P_{\mathcal{X}}$ denotes projection on the set $\mathcal{X} \subset \mathbb{R}^d$.

IG with diminishing stepsizes converges at rate $\mathcal{O}(1/\sqrt{k})$ for strongly convex sum function (Nedić & Bertsekas, 2001). If in addition to the strong convexity of the sum function, every component function $f_i$ is smooth, IG with diminishing stepsizes $\alpha_k = \Theta(1/k^s), s \in (0, 1]$ converges at rate $\mathcal{O}(1/k^s)$ (Gürbüzbalaban et al., 2015).

The convergence rate analysis of IG is valid regardless of order of processing the elements. However, in practice, the convergence rate of IG is known to be quite sensitive to the order of processing the functions (Bertsekas & Scientific, 2015; Gurbuzbalaban et al., 2017). If problem-specific knowledge can be used to find a favorable order $\sigma$ (defined as a permutation $\{\sigma_1, \cdots, \sigma_n\}$ of $\{1, 2, ..., n\}$), IG can be updated to process the functions according to this order, i.e.,

$$x_i^k = x_{i-1}^k - \alpha_k \nabla f_{\sigma_i}(x_{i-1}^k), \qquad i = 1, 2, \cdots, n. \tag{16}$$

In general a favorable order is not known in advance, and a common approach is sampling the function indices with replacement from the set $\{1, 2, \cdots, n\}$ and is called the Stochastic Gradient Descent (SGD) method, a.k.a. the Robbins-Monro algorithm (Robbins & Monro, 1951) (also see (Bottou, 1998; Bertsekas, 2015; Nemirovski et al., 2009; Shalev-Shwartz & Srebro, 2008)).

## 4.2 Convergence Rate of IG on Subsets Found by CRAIG

Next we analyze the convergence rate of IG applied to the weighted and ordered subset $S$ found by CRAIG. In particular, we show that (1) applying IG to $S$ converges to a close neighborhood of the optimal solution and that (2) this convergence happens at the same rate (same number of epochs) as IG on the full data. Formally, every step of IG on the subset becomes

$$x_i^k = x_{i-1}^k - \alpha_k \gamma_{s_{\sigma_i}} \nabla f_{s_{\sigma_i}}(x_{i-1}^k), \qquad i = 1, 2, \cdots, r, \quad s_i \in S, \quad |S| = r. \tag{17}$$

Here, $\sigma$ is a permutation of $\{1, 2, \cdots, r\}$, and the per-element stepsize $\gamma_{s_i}$ for every function $f_{s_i}$ is the weight of the element $s_i \in S$ and is fixed for all epochs.

## 4.3 Convergence Rate for Strongly Convex Functions

We first provide the convergence analysis for the case where the sum function in Problem (1) is strongly convex, i.e. $\forall x, y \in \mathbb{R}^d$ we have that $f(y) \geq f(x) + \langle \nabla f(x), y - x \rangle + \frac{\mu}{2}\|x - y\|^2$.

**Theorem 1.** *Assume that $\sum_{i \in V} f_i(x)$ is strongly convex, and $S$ is a weighted subset of size $r$ such that $L(S) = \sum_{i \in V} \min_{j \in S} d_{ij} \leq \epsilon$. Then for the iterates $\{x_k = x_0^k\}$ generated by applying IG to $S$ with per-epoch stepsize $\alpha_k = \alpha/k^\tau$ with $\alpha > 0$ and $\tau \in [0, 1]$, we have*

    *(i) if $\tau = 1$, then $\|x_k - x_*\|^2 \leq 2\epsilon R/\mu + r^2 \gamma_{\max}^2 C^2/k\mu$,*

    *(ii) if $0 < \tau < 1$, then $\|x_k - x_*\|^2 \leq 2\epsilon R/\mu$,*                                    *for $k \to \infty$*

    *(iii) if $\tau = 0$, then $\|x_k - x_*\|^2 \leq (1 - \alpha\mu)^{k+1}\|x_0 - x_*\|^2 + 2\epsilon R/\mu + \alpha r^2 \gamma_{\max}^2 C^2/\mu$,*

*where $C$ is an upper-bound on the norm of the component function gradients, i.e. $\max_{i \in V} \sup_{x \in \mathcal{X}} \|\nabla f_i(x)\| \leq C$, $\gamma_{\max} = \max_{j \in S} \gamma_j$ is the largest per-element step size, and $R = \min\{d_0, (r\gamma_{\max}C + \epsilon)/\mu\}$, where $d_0 = \|x_0 - x_*\|$ is the initial distance to the optimum $x_*$.*

All the proofs can be found in the Appendix. The above theorem shows that IG on $S$ converges at the same rate $\mathcal{O}(1/\sqrt{k})$ of IG on the entire data set $V$. However, compared to IG on $V$, the $|V|/|S|$ speedup of IG on $S$ comes at the price of getting an extra error term, $2\epsilon R/\mu$.

## 4.4 Convergence Rate for Smooth and Strongly Convex Functions

If in addition to strong convexity of the expected risk, each component function has a Lipschitz gradient, i.e. $\forall x \in \mathcal{X}, i \in [n]$ we have $\|\nabla f_i(x) - \nabla f_i(y)\| \leq \beta_i \|x - y\|$, then we get the following results about the iterates generated by applying IG to the weighted subset $S$ returned by CRAIG.

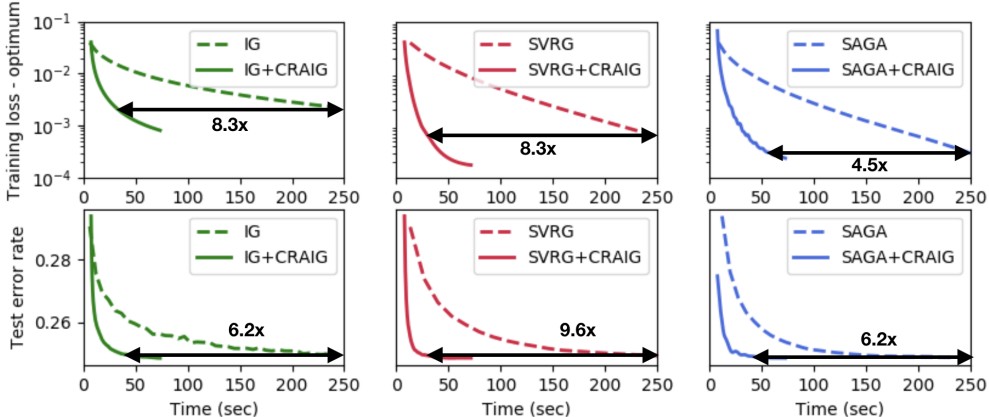

Figure 1: Loss residual and error rate of IG, SVRG, SAGA for Logistic Regression on Covtype data set with 581,012 data points. We compare performance of CRAIG (10% selected subset) vs. entire data set. We achieve the average speedup of 7x for achieving similar loss residual and error rate across the three optimization methods.

**Theorem 2.** *Assume that $\sum_{i \in V} f_i(x)$ is strongly convex and let $f_i(x), i = 1, 2, \cdots, n$ be convex and twice continuously differentiable component functions with Lipschitz gradients on $\mathcal{X}$. Given a subset $S$ such that $L(S) = \sum_{i \in V} \min_{j \in S} d_{ij} \leq \epsilon$. Then for the iterates $\{x_k = x_0^k\}$ generated by applying IG to $S$ with per-epoch stepsize $\alpha_k = \alpha/k^\tau$ with $\alpha > 0$ and $\tau \in [0, 1]$, we have*

    *(i) if $\tau = 1$, then $\|x_k - x_*\| \leq 2\epsilon/\mu + \beta C r \gamma_{\max}^2/k\mu$,*

    *(ii) if $0 < \tau < 1$, then $\|x_k - x_*\| \leq 2\epsilon/\mu$,*                      *for $k \to \infty$,*

    *(iii) if $\tau = 0$, then $\|x_k - x_*\| \leq (1 - \alpha\mu)^k \|x_0 - x_*\| + 2\epsilon/\mu + \alpha\beta C r \gamma_{\max}^2/\mu$,*

*where $\beta = \sum_{i=1}^n \beta_i$ is the sum of gradient Lipschitz constants of the component functions.*

The above theorem shows that for $\tau > 0$, IG applied to $S$ converges to a $2\epsilon/\mu$ neighborhood of the optimal solution, with a rate of $\mathcal{O}(1/k^\tau)$ which is the same convergence rate for IG on the entire data set $V$. As shown in our experiments, in real data sets small weighted subsets constructed by CRAIG provide a close approximation to the full gradient. Hence, applying IG to the weighted subsets returned by CRAIG provides a solution of the same or higher quality compared to the solution obtained by applying IG to the whole data set, in a considerably shorter amount of time.

## 5 EXPERIMENTS

In our experimental evaluation we wish to address the following questions: (1) How do loss and accuracy of IG applied to the subsets returned by CRAIG compare to loss and accuracy of IG applied to the entire data; (2) How small is the size of the subsets that we can select with CRAIG and still get a comparable performance to IG applied to the entire data; (3) how does the ordering affect the performance of IG on the subset; and (4) how well does CRAIG scale to large data sets, and extend to non-convex problems. To this end, we apply CRAIG to several convex and non-convex problems. In our experiments, we report the run-time as the wall-clock time for subset selection with CRAIG, plus minimizing the loss using IG or other optimizers with the specified learning rates. For the classification problems, we separately select subsets from each class while maintaining the class ratios in the whole data, and apply IG to the union of the subsets.

### 5.1 CONVEX EXPERIMENTS

In our convex experiments, we apply CRAIG to IG, as well as variance reduction methods SVRG (Johnson & Zhang, 2013), and SAGA (Defazio et al., 2014), that try to reduce the variance of SGD either based on computations of full gradients at pivot points, or on keeping per data point corrections

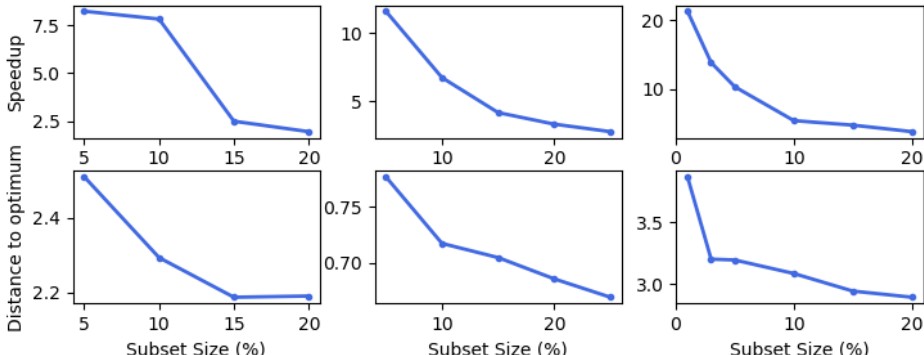

Figure 2: (top) Speedup of CRAIG applied to to get similar loss residual as IG after 50 epoch, and (bottom) distance to the optimal solution vs various subset sizes for (left) Covtype, (middle) SensIT, and (right) Ijcnn1. Smaller subsets provides larger speedups, but may converge farther away from the optimal solution.

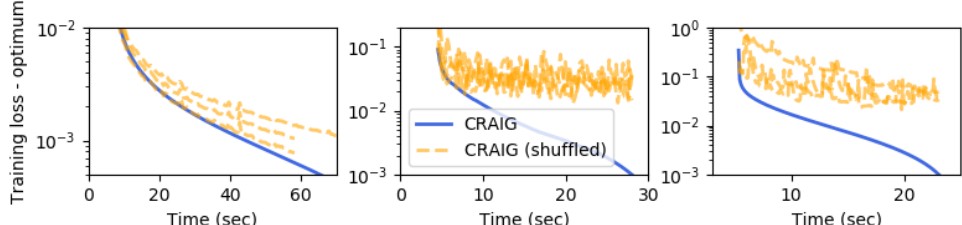

Figure 3: Loss residual of CRAIG for Logistic Regression on (left) Covtype, (middle) SensIT, and (right) Ijcnn, where we process the points of the subset according to the ordering provided by CRAIG vs three random permutations of the same subset. Notice convergence is significantly faster when we process the points in CRAIG order.

in memory. We apply L2-regularized logistic regression: $f_i(x) = \ln(1 + \exp(-x^T a_i y_i)) + 0.5\lambda x^T x$ to classify the following three datasets from LIBSVM: (1) *covtype.binary* including 581,012 data points of 54 dimensions, (2) *SensIT* including 78,823 training and 19,705 test data points of dimension 100, and (3) *Ijcnn1* including 49,990 training and 91,701 test data points of 22 dimensions. As *covtype* do not come with labeled test data, we randomly split the training data into halves to make the training/test split (training and set sets are consistent for different methods). For the convex experiments, $\lambda$ is set to $10^{-5}$.

Figure 1 compares training loss residual and test error rate of IG, SVRG, and SAGA on the subsets of size 10% of *covtype* selected by CRAIG (with corresponding per-element stepsizes) to that of IG, SVRG, and SAGA on the entire data set. We used a constant learning rate of $\alpha$ for SAGA and SVRG, and $\alpha/\sqrt{k}$ for IG, where $\alpha = 10^{-3}$ for Covtype and Ijcnn, and $3 \times 10^{-5}$ for SensIT. It can be seen that subsets obtained by CRAIG achieve a similar loss and error rate as the entire data sets, but much faster. In particular, we obtained a speedup of 8.3x, 8.3x, 4.5x from applying IG, SVRG and SAGA on the subsets of size 10% obtained by CRAIG.

Figure 2 top row, compares the speedup achieved by CRAIG to reach a similar loss residual as that of IG after 50 epochs for subsets of size 5% to 25%. The bottom row compares the L2-norm of the distance to optimal solution (estimated by running IG for a long time) for IG applied to the subsets of various size obtained by CRAIG. We observe that while smaller subsets provide a larger speedup, IG on smaller subsets may converge to a point farther away from the optimal solution.

Finally, Figure 3 shows the loss residual vs time for IG when it processes the elements of the subsets according to the ordering obtained by CRAIG compared to random permutations of the same subsets. We observe that the greedy ordering significantly improves the rate of convergence of IG.

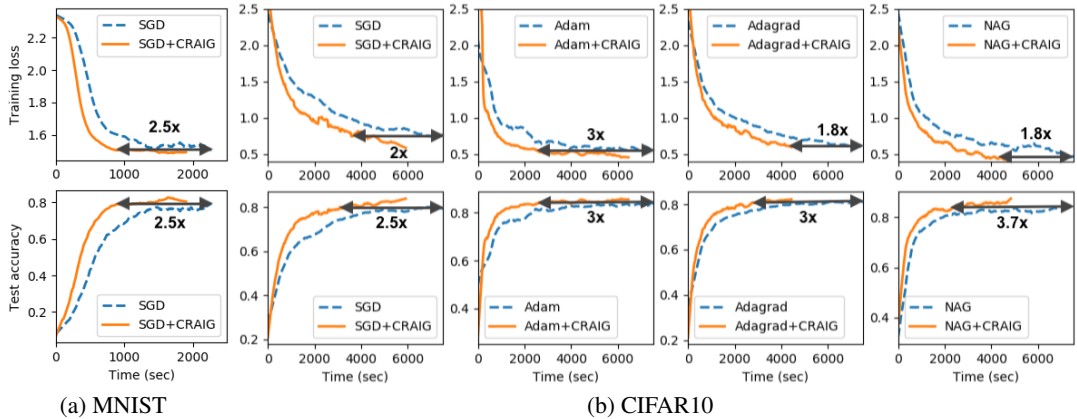

(a) MNIST                  (b) CIFAR10

Figure 4: Training loss and test accuracy of CRAIG applied to (a) SGD on MNIST with a 1-layer neural network, and (b) SGD, Adam, Adagrad, NAG, on CIFAR-10 with ResNet-56. CRAIG provides 2x to 3x speedup and a better generalization performance.

## 5.2 NON-CONVEX EXPERIMENTS

Our non-convex experiments involves applying CRAIG to train the following two neural networks: (1) Our smaller network is a fully-connected hidden layer of 100 nodes and ten softmax output nodes; sigmoid activation and L2 regularization with $\lambda = 0.0001$ and mini-batches of size 10 on MNIST dataset of handwritten digits containing 60,000 training and 10,000 test images. (2) Our large neural network is ResNet-56 for CIFAR10 with convolution, average pooling and dense layers with softmax outputs and L2 regularization with $\lambda = 2 \times 10^{-4}$ CIFAR 10 includes 50,000 training and 10,000 test images from 10 classes, and we used mini-batches of size 128. Both MNIST and CIFAR10 data sets are normalized into [0, 1] by division with 255.

We apply CRAIG to several popular methods for training neural networks, including SGD, Nesterov Accelerated Gradient (NAG) method (Nesterov, 1983), Adagrad (Duchi et al., 2011), and Adaptive Moment Estimation (Adam) (Kingma & Ba, 2014). Momentum is a method that helps accelerate SGD in the relevant direction and dampens oscillations. NAG adapts momentum updates to the slope of our error function and speed up SGD. Adagrad adapts the learning rate to the parameters, performing smaller updates (i.e. low learning rates) for parameters associated with frequently occurring features, and larger updates (i.e. high learning rates) for parameters associated with infrequent features. Adam computes individual adaptive learning rates for different parameters from estimates of first and second moments of the gradients. Fig. 4a shows loss, and error rate for training a 1-layer neural net on MNIST. Fig. 4b show similar quantities for training ResNet-56 on CIFAR10. For both problems, we used a constant learning rate of $10^{-2}$. Here, we apply CRAIG to select a subset of 30%-40% of the data at the beginning of every epoch and train only on the selected subset with the corresponding per-element stepsizes. Interestingly, in addition to achieving a speedup of 2x to 3x for training neural networks, the subsets selected by CRAIG provide a better generalization performance compared to models trained on the entire data set.

## 6 CONCLUSION

We developed a method, CRAIG, for selecting an ordered subset (coreset) of data points with their corresponding per-element stepsizes to speed up iterative gradient (IG) methods. In particular, we showed that weighted subsets that minimize the upper-bound on the estimation error of the full gradient, maximize a submodular facility location function. Hence, we can obtain an ordered subset of data points with their corresponding learning rates using a greedy algorithm. We showed that IG on subsets $S$ returned by CRAIG converges at the same rate as IG on the entire data set $V$, while providing a $|V|/|S|$ speedup. In our set of experiments, we showed that various IG methods, including SAGA, SVRG, NAG, Adagrad and Adam runs up to 10x faster on convex and up to 3x on non-convex problems on subsets found by CRAIG while achieving practically the same trining loss and test error. Finally, we empirically demonstrated the effect of the ordering found by the greedy algorithm on the convergence rate of IG methods.

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

## A   CONVERGENCE RATE ANALYSIS

We firs proof the following Lemma which is an extension of the [Chung et al. (1954), Lemma 4].

**Lemma 3.**  *Let $u_k \geq 0$ be a sequence of real numbers. Assume there exist $k_0$ such that*

$$u_{k+1} \leq (1 - \frac{c}{k})u_k + \frac{e}{k^p} + \frac{d}{k^{p+1}}, \quad \forall k \geq k_0,$$

*where $e > 0, d > 0, c > 0$ are given real numbers. Then*

$$u_k \leq (dk^{-1} + e)(c - p + 1)^{-1}k^{-p+1} + o(k^{-p+1}) \qquad \text{for } c > p - 1, p \geq 1 \qquad (18)$$

$$u_k = O(k^{-c}\log k) \qquad \text{for } c = p - 1, p > 1 \qquad (19)$$

$$u_k = O(k^{-c}) \qquad \text{for } c < p - 1, p > 1 \qquad (20)$$

$$\qquad (21)$$

*Proof.* Let $c > p - 1$ and $v_k = k^{p-1}u_k - \frac{d}{k(c-p+1)} - \frac{e}{c-p+1}$. Then, using Taylor approximation $(1 + \frac{1}{k})^p = (1 + \frac{p}{k}) + o(\frac{1}{k})$ we can write

$$v_{k+1} = (k+1)^{p-1}u_{k+1} - \frac{d}{(k+1)(c-p+1)} - \frac{e}{c-p+1} \qquad (22)$$

$$\leq k^{p-1}(1 + \frac{1}{k})^{p-1}\left((1 - \frac{c}{k})u_k + \frac{e}{k^p} + \frac{d}{k^{p+1}}\right) - \frac{d}{(k+1)(c-p+1)} - \frac{e}{c-p+1} \qquad (23)$$

$$= k^{p-1}u_k\left(1 - \frac{c-p+1}{k} + o(\frac{1}{k})\right) + \frac{e}{k}\left(1 + \frac{p-1}{k} + o(\frac{1}{k})\right) \qquad (24)$$

$$+ \frac{d}{k^2}\left(1 + \frac{p-1}{k} + o(\frac{1}{k})\right) - \frac{d}{(k+1)(c-p+1)} - \frac{e}{c-p+1} \qquad (25)$$

$$= \left(v_k + \frac{d}{k(c-p+1)} + \frac{e}{c-p+1}\right)\left(1 - \frac{c-p+1}{k} + o(\frac{1}{k})\right) \qquad (26)$$

$$+ \frac{e}{k}\left(1 + \frac{p-1}{k} + o(\frac{1}{k})\right) + \frac{d}{k^2}\left(1 + \frac{p-1}{k} + o(\frac{1}{k})\right) \qquad (27)$$

$$- \frac{d}{(k+1)(c-p+1)} - \frac{e}{c-p+1} \qquad (28)$$

$$= v_k\left(1 - \frac{c-p+1}{k} + o(\frac{1}{k})\right) + \frac{d/(c-p+1)}{k(k+1)} + \frac{e(p-1)}{k^2} + \frac{d(p-1)}{k^3} + o(\frac{1}{k^2}) \qquad (29)$$

Note that for $v_k$, we have

$$\sum_{k=0}^{\infty}\left(1 - \frac{c-p+1}{k} + o(\frac{1}{k})\right) = \infty$$

and

$$\left(\frac{d/(c-p+1)}{k(k+1)} + \frac{e(p-1)}{k^2} + \frac{d(p-1)}{k^3} + o(\frac{1}{k^2})\right)\left(1 - \frac{c-p+1}{k} + o(\frac{1}{k})\right)^{-1} \to 0.$$

Therefore, $\lim_{k\to\infty} v_k \leq 0$, and we get Eq. 18. For $p = 1$, we have $u_k \leq \frac{e}{c}$. Hence, $u_k$ converges into the region $u \leq \frac{e}{c}$, with ratio $1 - \frac{c}{k}$.

Moreover, for $p - 1 \geq c$ we have

$$v_{k+1} = u_{k+1}(k+1)^c \leq \left[(1 - \frac{c}{k})u_k + \frac{e}{k^p} + \frac{d}{k^{p+1}}\right]k^c\left(1 + \frac{c}{k} + \frac{c^2}{2k^2} + o(\frac{1}{k^2})\right) \qquad (30)$$

$$= \left(1 - \frac{c^2}{2k^2} + o(\frac{1}{k^2})\right)v_k + \frac{d}{k^{p-c+1}}\left(1 + O(\frac{1}{k})\right) + \frac{e}{k^{p-c}}\left(1 + \frac{c}{k} + O(\frac{1}{k^2})\right) \qquad (31)$$

$$\leq v_k + \frac{e'}{k^{p-c}} \qquad (32)$$

for sufficiently large $k$. Summing over $k$, we obtain that $v_k$ is bounded for $p - 1 > c$ (since the series $\sum_{k=1}^{\infty}(1/k^{\alpha})$ converges for $\alpha > 1$) and $v_k = O(\log k)$ for $p = c + 1$ (since $\sum_{i=1}^{k}(1/i) = O(\log k)$). $\qquad \square$

In addition, based on [Chung et al. (1954), Lemma 5] for $u_k \geq 0$, we can write

$$u_{k+1} \leq (1 - \frac{c}{k^s})u_k + \frac{e}{k^p} + \frac{d}{k^t}, \qquad 0 < s < 1, s \leq p < t. \tag{33}$$

Then, we have

$$u_k \leq \frac{e}{c}\frac{1}{k^{p-s}} + o(\frac{1}{k^{p-s}}). \tag{34}$$

### A.1 CONVERGENCE RATE FOR STRONGLY CONVEX FUNCTIONS

PROOF OF THEOREM 1

We now provide the convergence rate for strongly convex functions building on the analysis of Nedić & Bertsekas (2001). For non-smooth functions, gradients can be replaced by sub-gradients.

Let $x_k = x_0^k$. For every IG update on subset $S$ we have

$$\|x_j^k - x_*\|^2 = \|x_{j-1}^k - \alpha_k\gamma_j\nabla f_j(x_{j-1}^k) - x_*\|^2 \tag{35}$$

$$= \|x_{j-1}^k - x_*\|^2 - 2\alpha_k\gamma_j\nabla f_j(x_{j-1}^k)(x_{j-1}^k - x_*) + \alpha_k^2\|\gamma_j\nabla f_j(x_{j-1}^k)\|^2 \tag{36}$$

$$\leq \|x_{j-1}^k - x_*\|^2 - 2\alpha_k(f_j(x_{j-1}^k) - f_i(x_*)) + \alpha_k^2\|\gamma_i\nabla f_j(x_{j-1}^k)\|^2. \tag{37}$$

Adding the above inequalities over elements of $S$ we get

$$\|x_{k+1} - x_*\|^2 \leq \|x_k - x_*\|^2 - 2\alpha_k\sum_{j\in S}(f_i(x_{j-1}^k) - f_j(x_*)) + \alpha_k^2\sum_{j\in S}\|\gamma_j\nabla f_j(x_{j-1}^k)\|^2 \tag{38}$$

$$= \|x_k - x_*\|^2 - 2\alpha_k\sum_{j\in S}(f_j(x_k) - f_i(x_*))$$

$$+ 2\alpha_k\sum_{j\in S}(f_j(x_{j-1}^k) - f_j(x_k)) + \alpha_k^2\sum_{j\in S}\|\gamma_j\nabla f_j(x_{j-1}^k)\|^2 \tag{39}$$

Using strong convexity we can write

$$\|x_{k+1} - x_*\|^2 \leq \|x_k - x_*\|^2 - 2\alpha_k\Big(\sum_{j\in S}\gamma_j\nabla f_j(x_*)\cdot(x_k - x_*) + \frac{\mu}{2}\|x_k - x_*\|^2\Big)$$

$$+ 2\alpha_k\sum_{j\in S}(f_j(x_{j-1}^k) - f_j(x_k)) + \alpha_k^2\sum_{j\in S}\|\gamma_j\nabla f_j(x_{j-1}^k)\|^2 \tag{40}$$

Using Cauchy–Schwarz inequality, we know

$$|\sum_{j\in S}\gamma_j\nabla f_j(x_*)\cdot(x_k - x_*)| \leq \|\sum_{j\in S}\gamma_j\nabla f_j(x_*)\|\cdot\|x_k - x_*\|. \tag{41}$$

Hence,

$$-\sum_{j\in S}\gamma_j\nabla f_j(x_*)\cdot(x_k - x_*) \leq \|\sum_{j\in S}\gamma_j\nabla f_j(x_*)\|\cdot\|x_k - x_*\|. \tag{42}$$

From reverse triangle inequality, and the facts that $S$ is chosen in a way that $\|\sum_{i\in V}\nabla f_i(x_*) - \sum_{j\in S}\gamma_j\nabla f_j(x_*)\| \leq \epsilon$, and that $\sum_{i\in V}\nabla f_i(x_*) = 0$ we have $\|\sum_{j\in S}\gamma_j\nabla f_j(x_*)\| \leq \|\sum_{i\in V}\nabla f_i(x_*)\| + \epsilon = \epsilon$. Therefore

$$\|\sum_{j\in S}\gamma_j\nabla f_j(x_*)\|\cdot\|x_k - x_*\| \leq \epsilon\cdot\|x_k - x_*\| \tag{43}$$

For a continuously differentiable function, the following condition is implied by strong convexity condition

$$\|x_k - x_*\| \leq \frac{1}{\mu}\|\sum_{j\in S}\gamma_j\nabla f_j(x_k)\|. \tag{44}$$

Assuming gradients have a bounded norm $\max_{\substack{x \in \mathcal{X} \\ j \in V}} \|\nabla f_j(x)\| \leq C$, and the fact that $\sum_{j \in S} \gamma_j = n$ we can write

$$\|\sum_{j \in S} \gamma_j \nabla f_j(x_k)\| \leq n \cdot C. \tag{45}$$

Thus for initial distance $\|x_0 - x_*\| = d_0$, we have

$$\|x_k - x_*\| \leq \min(n \cdot C, d_0) = R \tag{46}$$

Putting Eq. 42 to Eq. 46 together we get

$$\|x_{k+1} - x_*\|^2 \leq (1 - \alpha_k \mu)\|x_k - x_*\|^2 + 2\alpha_k \epsilon R/\mu$$
$$+ 2\alpha_k \sum_{j \in S}(f_j(x_{j-1,k}) - f_j(x_k)) + \alpha_k^2 r \gamma_{\max}^2 C^2. \tag{47}$$

Now, from convexity of every $f_j$ for $j \in S$ we have that

$$f_j(x_k) - f_j(x_{j-1}^k) \leq \|\gamma_j \nabla f_j(x_k)\| \cdot \|x_{j-1}^k - x_k\|. \tag{48}$$

In addition, incremental updates gives us

$$\|x_{j-1}^k - x_k\| \leq \alpha_k \sum_{i=1}^{j-1} \|\gamma_i \nabla f_i(x_{i-1}^k)\| \leq \alpha_k(j-1)\gamma_{\max}C. \tag{49}$$

Therefore, we get

$$2\alpha_k \sum_{j \in S}(f_j(x_k) - f_j(x_{j-1}^k)) + \alpha_k^2 r \gamma_{\max}^2 C^2$$

$$\leq 2\alpha_k \sum_{i=1}^{r} \gamma_{\max}C \cdot \alpha_k(j-1)\gamma_{\max}C + \alpha_k^2 r \gamma_{\max}^2 C^2 \tag{50}$$

$$= \alpha_k^2 r^2 \gamma_{\max}^2 C^2 \tag{51}$$

Hence,

$$\|x_{k+1} - x_*\|^2 \leq (1 - \alpha_k \mu)\|x_k - x_*\|^2 + 2\alpha_k \epsilon R/\mu + \alpha_k^2 r^2 \gamma_{\max}^2 C^2. \tag{52}$$

where $\gamma_{\max}$ is the size of the largest cluster, and $C$ is the upperbound on the gradients.

For $0 < s \leq 1$, the theorem follows by applying Lemma 3 to Eq. 52, with $c = \mu$, $e = 2\epsilon R/\mu$, and $d = r^2 \gamma_{\max}^2 C^2$.

For $s = 0$, where we have a constant step size $\alpha_k = \alpha \leq \frac{1}{\mu}$, we get

$$\|x_{k+1} - x_*\|^2 \leq (1 - \alpha\mu)^{k+1}\|x_0 - x_*\|^2$$

$$+ 2\epsilon\alpha\epsilon R \sum_{j=0}^{k}(1 - \alpha\mu)^j/\mu + \alpha^2 r^2 \gamma_{\max}^2 C^2 \sum_{j=0}^{k}(1 - \alpha\mu)^j \tag{53}$$

Since $\sum_{j=0}^{k}(1 - \alpha\mu)^j \leq \frac{1}{\alpha\mu}$, we get

$$\|x_{k+1} - x_*\|^2 \leq (1 - \alpha\mu)^{k+1}\|x_0 - x_*\|^2 + 2\epsilon R/\mu + \alpha r^2 \gamma_{\max}^2 C^2/\mu, \tag{54}$$

and therefore,

$$\|x_{k+1} - x_*\|^2 \leq (1 - \alpha\mu)^{k+1}\|x_k - x_*\|^2 + 2\epsilon R/\mu + \alpha r^2 \gamma_{\max}^2 C^2/\mu. \tag{55}$$

## A.2 Convergence Strongly Convex and Smooth Component Functions

### Proof of Theorem 2

IG updates for cycle $k$ on subset $S$ can be written as

$$x_{k+1} = x_k - \alpha_k(\sum_{j \in S} \gamma_j \nabla f_j(x_k) - e_k) \tag{56}$$

$$e_k = \sum_{j \in S} \gamma_i(\nabla f_j(x_k) - \nabla f_j(x_{j-1}^k)) \tag{57}$$

Building on the analysis of Gürbüzbalaban et al. (2015), for convex and twice continuously differentiable function, we can write

$$\sum_{j \in S} \gamma_j \nabla f_j(x_k) - \sum_{j \in S} \gamma_j \nabla f_j(x_*) = A_k^r(x_k - x_*) \tag{58}$$

where $A_k^r = \int_0^1 \nabla^2 f(x_* + \tau(x_k - x_*))d\tau$ is average of the Hessian matrices corresponding to the $r$ (weighted) elements of $S$ on the interval $[x_k, x_*]$.

From Eq. 58 we have

$$\sum_{i \in V}(\nabla f_i(x_k) - \nabla f_i(x_*)) - \sum_{j \in S}\gamma_j(\nabla f_j(x_k) - \nabla f_j(x_*)) = A_k(x_k - x_*) - A_k^r(x_k - x_*), \tag{59}$$

where $A_k$ is average of the Hessian matrices corresponding to all the $n$ component functions on the interval $[x_k, x_*]$. Taking norm of both sides and noting that $\sum_{i \in V} f_i(x_*) = 0$ and hence $\|\sum_{j \in S} \gamma_j f_j(x_*)\| \leq \epsilon$, we get

$$\|(A_k - A_k^r)(x_k - x_*)\| = \left\|\left(\sum_{i \in V}\nabla f_i(x_k) - \sum_{j \in S}\gamma_j \nabla f_j(x_k)\right) + \sum_{j \in S}\gamma_j f_j(x_*)\right\| \leq 2\epsilon, \tag{60}$$

where $\epsilon$ is the estimation error of the full gradient by the weighted gradients of the elements of the subset $S$, and we used $\|\sum_{i \in V}\nabla f_i(x_k) - \sum_{j \in S}\gamma_j \nabla f_j(x_k)\| \leq \epsilon$.

Substituting Eq. 58 into Eq. 56 we obtain

$$x_{k+1} - x_* = (I - \alpha_k A_k^r)(x_k - x_*) + \alpha_k e_k \tag{61}$$

Taking norms of both sides, we get

$$\|x_{k+1} - x_*\| \leq \|(I - \alpha_k A_k^r)(x_k - x_*\|) + \alpha_k\|e_k\| \tag{62}$$

Now, we have

$$\|(I - \alpha_k A_k^r)(x_k - x_*)\| = \|I(x_k - x_*) - \alpha_k A_k^r(x_k - x_*)\| \tag{63}$$
$$= \|I(x_k - x_*) - \alpha_k(A_k^r - A_k)(x_k - x_*) - \alpha_k A_k(x_k - x_*)\| \tag{64}$$
$$\leq \|(I - \alpha_k A_k)(x_k - x_*)\| + \alpha_k\|(A_k - A_k^r)(x_k - x_*)\| \tag{65}$$
$$\leq \|(I - \alpha_k A_k)(x_k - x_*)\| + 2\alpha_k\epsilon \tag{66}$$

Substituting into Eq. 62, we obtain

$$\|x_{k+1} - x_*\| \leq \|I - \alpha_k A_k\| \cdot \|x_k - x_*\| + 2\alpha_k\epsilon + \alpha_k\|e_k\| \tag{67}$$

From strong convexity of $\sum_{i \in V} f_i(x)$, and gradient smoothness of each component $f_i(x)$ we have

$$\mu I_n \preceq \sum_{i \in V}\nabla^2 f_i(x), A_k \preceq \beta I_n, \quad x \in \mathcal{X}, \tag{68}$$

where $\beta = \sum_{i \in V}\beta_i$ In addition, from the gradient smoothness of the components we can write

$$\|e_k\| \leq \sum_{j \in S}\gamma_j\beta_j\|x_k - x_j^k\| \tag{69}$$

$$\leq \sum_{j \in S}\gamma_j\beta_j\sum_{i=1}^{j-1}\|x_{i-1}^k - x_i^k\| \tag{70}$$

$$\leq \sum_{j \in S}\gamma_j\beta_j\alpha_k\sum_{i=1}^{j-1}\|\gamma_i\nabla f_i(x_i^k)\| \tag{71}$$

$$\leq \alpha_k\beta Cr\gamma_{\max}^2, \tag{72}$$

where in the last line we used $|S| = r$. Therefore,

$$\|x_{k+1} - x_*\| \leq \max(\|1 - \alpha_k \mu\|, \|1 - \alpha_k \beta\|) \|x_k - x_*\| + 2\alpha_k \epsilon + \alpha_k^2 \beta Cr\gamma_{\max}^2 \qquad (73)$$

$$\leq (1 - \alpha_k \mu)\|x_k - x_*\| + 2\alpha_k \epsilon + \alpha_k^2 \beta Cr\gamma_{\max}^2 \quad \text{if} \quad \alpha_k \beta \leq 1. \qquad (74)$$

For $0 < s \leq 1$, the theorem follows by applying Lemma 3 to Eq. 73 with $c = \mu$, $e = 2\epsilon$, $d = \beta Cr\gamma_{\max}^2$. For $s = 0$, where we have a constant step size $\alpha_k = \alpha \leq \frac{1}{\beta}$, we get

$$\|x_{k+1} - x_*\| \leq (1 - \alpha\mu)^{k+1}\|x_k - x_*\| + 2\epsilon \sum_{i=0}^{k}(1 - \alpha\mu)^i + \alpha^2 \sum_{i=0}^{k}(1 - \alpha\mu)^i \beta Cr\gamma_{\max}^2 \qquad (75)$$

$$\leq (1 - \alpha\mu)^{k+1}\|x_k - x_*\| + 2\epsilon/\mu + \alpha\beta Cr\gamma_{\max}^2/\mu, \qquad (76)$$

$$\leq (1 - \alpha\mu)^{k+1}\|x_k - x_*\| + 2\epsilon/\mu + Cr\gamma_{\max}^2/\mu, \qquad (77)$$

where the inequality in Eq. 76 follows since $\sum_{i=0}^{k}(1 - \alpha\mu)^i \leq \frac{1}{\alpha\mu}$.

## B  NORM OF THE DIFFERENCE BETWEEN GRADIENTS

For ridge regression $f_i(x) = \frac{1}{2}(\langle a_i, x \rangle - y_i)^2 + \frac{\lambda}{2}\|x\|^2$, we have $\nabla f_i(x) = a_i(\langle a_i, x \rangle - y_i) + \lambda x$. Therefore,

$$\|\nabla f_i(x) - \nabla f_j(x)\| = (\|a_i - a_j\|.\|x\| + \|y_i - y_j\|)\|a_j\| \qquad (78)$$

For $\|a_i\| \leq 1$, and $y_i = y_j$ we get

$$\|\nabla f_i(x) - \nabla f_j(x)\| \leq \|a_i - a_j\| O(\|x\|) \qquad (79)$$

For regularized logistic regression with $y \in \{-1, 1\}$, we have $\nabla f_i(x) = y_i/(1 + e^{y_i \langle a_i, x \rangle})$. For $y_i = y_j$ we get

$$\|\nabla f_i(x) - \nabla f_j(x)\| = \frac{e^{\|a_i - a_j\|.\|x\|} - 1}{1 + e^{-\langle a_i, x \rangle}}\|a_j\|. \qquad (80)$$

For $\|a_i\| \leq 1$, using Taylor approximation $e^x \leq 1 + x$, and noting that $\frac{1}{1 + e^{-\langle a_i, x \rangle}} \leq 1$ we get

$$\|\nabla f_i(x) - \nabla f_j(x)\| \leq \frac{\|a_i - a_j\|.\|x\|}{1 + e^{-\langle a_i, x \rangle}}\|a_j\| \leq \|a_i - a_j\| O(\|x\|). \qquad (81)$$

For classification, we require $y_i = y_j$, hence we can select subsets from each class and then merge the results. On the other hand, in ridge regression we also need $|y_i - y_j|$ to be small. Similar results can be deduced for other loss functions including square loss, smoothed hinge loss, etc.

Assuming $\|x\|$ is bounded for all $x \in \mathcal{X}$, upper-bounds on the euclidean distances between the gradients can be pre-computed.

## C  ADDITIONAL EXPERIMENTS

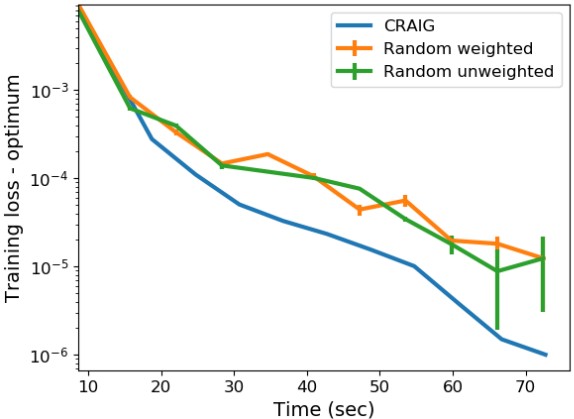

Figure 5: Loss residual vs. time for IG on 10 subsets of size 10%, 20%, 30%, ..., 100% selected by CRAIG, random subsets, and random subsets weighted by $|V|/|S|$. Stepsizes are tuned for every subset separately, by preferring smaller training loss from a large number of parameter combinations for two types of learning scheduling: exponential decay $\eta(t) = \eta_0 a^{\lfloor t/n \rfloor}$ with parameters $\eta_0$ and $a$ to adjust and $t$-inverse $\eta(t) = \eta_0(1 + b\lfloor t/n \rfloor)^{-1}$ with $\eta_0$ and $b$ to adjust.

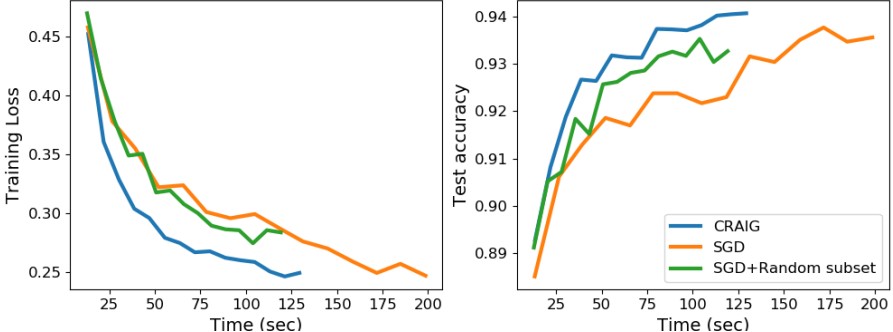

Figure 6: Training loss and test accuracy for SGD applied to full MNIST vs. subsets of size 60% selected by CRAIG and random subsets of size 60%. Both the random subsets and the subsets found by CRAIG change at the beginning of every epoch.

