# OpenReview forum: "Coresets for Accelerating Incremental Gradient Methods"
_ICLR.cc/2020/Conference — Reject_

### Official Review · AnonReviewer3 · 2019-10-09
**Official Blind Review #3**

**Rating:** 8

**Review:**

This paper presents a method for subselecting training data
in order to speed up incremental gradient (IG) methods (in terms of computation time).
The idea is to train a model on a representative subset of the data such
that the computation time is significantly decreased without
significantly degrading performance. Given a dataset D and
subset S selected by the proposed method, it is shown that
training on subset S achieves nearly the same loss as training on the full
dataset D would, while achieving significant computational speedups.

This paper is a clear accept. The approach is novel and has well-developed
theory supporting it. The empirical evaluation of the method shows
large speedups in training time without degradation in performance
for reasonably large subsets (e.g. 20% of the data). The paper is
very clear, well-written, and was a genuinely fun read.

Clarifying questions:
  - In results reporting speedups, does the reported training time for CRAIG
  include the preprocessing time? Or only the time spent running IG on the resulting
  subset?
  - How many runs are the experiments averaged over? There don't seem to be
  error bars, which makes it difficult to assess whether the speedups are
  statistically significant
  - I imagine that an approach like this would be desirable when working with  very large datasets. Has
  CRAIG been evaluated in settings with millions of datapoints? Or does it become impractical? I think
  that the paper stands on its own without such a demonstration, but it would go a long way towards
  encouraging mainstream adoption of your method.
  - Figure 3, left: What could be happening at around 40s? It looks like
  all three of the random permutations have a spike in loss at around the same time, despite being
  different permutations of the sub-selected data
  - How were hyperparameters, such as the regularization parameter, step-size etc. chosen? One of the
  main claims of the paper is that using the subset selected by CRAIG doesn't significantly
  effect the optimization performance. But if the baselines weren't thoroughly tuned, it could be the case
  that IG on the CRAIG subset performs similarly to IG on the full training data, but that neither
  is actually reaching satisfactory performance in a given domain.
  - Figure 4: isn't 2000s \approx 30min really slow for MNIST? From what I remember, reasonable test accuracy
  on MNIST with a feed-forward network with a single layer takes only a few minutes? Though admittedly, I could
  be misremembering this.

Somewhat open-ended questions:
  - To what extent are the results hardware dependent? Do you see similar results on
  different hardware? I'm wondering how much of the speedup could be attributed to
  something like better memory locality when using the smaller subset selected using CRAIG.
  - Section 3.4 mentions that the O(|V||S|) complexity can be reduced
  to O(|V|) using a stochastic greedy algorithm. Has the performance
  when training on a subset selected via the stochastic algorithm
  been compared to the performance when training on a subset selected by the
  deterministic version?

I have only minor suggestions:
  - The CRAIG Algorithm
    - When F is introduced, I had trouble conceptualizing what kind of object it was. I think
      mentioning what spaces it's mapping between could increase readability.
    - Mirzasoluiman et al. (2015a) looks like it is supposed to be a parenthetical citation
  - Figure 4: The caption labels appear to be swapped. In the figure, (a) is MNIST, but in the
    caption, (b) is MNIST
  - 5.1: There is a vertical space issue between the introduction of section 5 and section 5.1.
  I suspect this was necessary to make the max page requirements. If space is an issue, my suggestion would
  be to move Algorithm 1 to the appendix. It's nice to have a concrete algorithm specification, but I personally
  did not find that it aided my understanding of your paper.

**Experience Assessment:**

I have read many papers in this area.

**Review Assessment: Checking Correctness Of Derivations And Theory:**

I assessed the sensibility of the derivations and theory.

**Review Assessment: Checking Correctness Of Experiments:**

I carefully checked the experiments.

**Review Assessment: Thoroughness In Paper Reading:**

I read the paper thoroughly.

---

> ### Author Response · Authors · 2019-11-15
> **Detailed Responses**
>
> We thank the reviewer for acknowledging the novelty and the theoretical strength of our work and for noting that our​ experiments ​are​ ​solid​ and our setup and analyses are sound.
>
> RE: Total running time of our approach
> All the run time results include the preprocessing time for logistic regression, and finding a different subset per epoch for neural network experiments.
>
> RE: Error bars
> The experiments report the average over 5 runs for IG and 1 run for CRAIG as it uses a deterministic order. We added error bars to Figure 5 in the appendix C and will add error bars to all the plots in the final version. The error bars in Figure 5 shows that the speedup achieved by CRAIG is significant.
>
> RE: Experiments on massive data
> The rich literature on large-scale submodular maximization allows to find the near-optimal subset efficiently, and as a result CRAIG is practical and scales well to large datasets. We are working on a large scale experiment on ImageNet to include in the final version of the manuscript.
>
> RE: Step-size and regularization
> We repeated our largest convex experiments with carefully tuned stepsizes for SGD when the subset is found by 1) CRAIG, 2) uniformly at random with weights equal to 1, and 3) uniformly at random with all the weights equal to $|V|/|S|$. Our results show the superior performance of CRAIG over random weighted and unweighted subsets. It can be observed that for carefully tuned stepsizes random subsets and random subsets upweighted by $|V|/|S|$ show similar performance. This confirms the quality of per-element stepsizes chosen by CRAIG. Although we do not report the tuning time in this Figure, we note that tuning the initial rate and the decay factor for learning rate of SGD is significantly faster on the subset found by CRAIG compared to the full dataset.
>
> RE: MNIST experiment
> We thank the reviewer for pointing out this inconsistency. Using a better learning rate and bigger batch, we indeed got the expected result. It can be seen in Figure 6 in the Appendix C that CRAIG while achieving the same training loss, has a considerably better generalization performance.
>
> RE: Covtype experiment around 40sec
> We are not sure about the pattern observed around 40sec. We will use better colors and line styles to improve the quality of the Figure.
>
> RE: Better memory locality
> This is a very interesting point. While we have not yet looked at the hardware dependent benefits resulted from using a smaller subset, it would be very beneficial to look more closely into such properties. This is particularly interesting for convex loss functions where the subset is found as a preprocessing step and eliminates the need for random access memories to various parts of a massive dataset.
>
> RE: Stochastic greedy
> In our experiments, we observed that we get a considerable speedup by using stochastic greedy instead of the classical greedy algorithm, without compromising the solution quality. We will add performance comparison with stochastic greedy to the final version of the manuscript.
>
> RE: Minor suggestions
> We thank the reviewer for pointing out the mislabeled subfigures. We modified them accordingly. We will add more explanation on the objective function F to improve readability.

---

### Official Review · AnonReviewer1 · 2019-10-20
**Official Blind Review #1**

**Rating:** 3

**Review:**

This paper proposes a novel extension to SGD/incremental gradient methods called CRAIG. The algorithm selects a subset of datapoints to approximate the training loss at the beginning of each epoch in order to reduce the total amount of time necessary to solve the empirical risk minimization problem. In particular, the algorithm formulates a submodular optimization problem based on the intuition that the gradient of the problem on the selected subset approximates the gradient of the true training loss up to some tolerance. Each datapoint in the subset is a medoid and assigned a weight corresponding to the number of datapoints in the full set that are assigned to that particular datapoint. A greedy algorithm is employed to approximately solve the subproblem. Theory is proven based on based on an incremental subgradient method with errors. Experiments demonstrate significant savings in time for training both logistic regression and small neural networks.

Strengths:

The proposed idea is novel and intriguing, utilizing tools from combinatorial optimization to select an appropriate subset for approximating the training loss. Based on the experiments provided in the paper, it does appear to yield a significant speedup in training time. It is interesting to observe how the order of the datapoints matter significantly for training, and that CRAIG is also able to naturally define a good ordering of the datapoints for SG training. This is strong algorithmic work.

Weaknesses:

Some questions I had about the work:

- How well does one have to approximate $d_{ij}$ in order for the method to be effective? The authors provide an approach to approximate this for both logistic regression and neural networks. How does one guarantee that one is obtaining the maximum over $x in \mathcal{X}$ for neural networks via backpropagating only on the last layer? Does taking this maximum matter?
- How does one choose $\epsilon$? Is this related to how $d_{ij}$’s are approximated?
- If one were to consider an algorithm that samples points from this new distribution over the data given by CRAIG, if one were to include the weight $\gamma_j$ into the algorithm, would the sample gradient be unbiased? What if one were to simply use $\gamma_j$ to weight that particular sample in the new distribution?
- In machine learning, the empirical risk (finite-sum) minimization problem is an approximation to the true expected risk minimization problem. What is the effect of CRAIG on the expected risk? Is there any deterioration in generalization performance?
- In page 4, what does the $\min_{S \subseteq V}$ refer to? Should the equation be interpreted as with the set $S$ fixed or not?
- Theorems 1 and 2 are stated a bit non-rigorously. Are these theorems for fixed $k$? What does it mean for these bounds that $k \rightarrow \infty$?
- In Theorems 1 and 2, what is the bound on the steplength in order to obtain the convergence result for $\tau = 0$?
- In Theorem 1 for $0 < \tau < 1$, why does one obtain a result where $\|x_k – x_*\|^2 \leq 2 \epsilon R / \mu$, why is the distance to the solution bounded by a constant? What if one were to initialize $x_0$ to be such that $\|x_0 – x_*\|^2 > 2 \epsilon R / \mu$? (Similar for Theorem 2.)
- In the experiments, how is the steplength and other hyperparameters tuned? Are multiple trials run?
- Is $\epsilon$ used to determine the subset or is it based on a predetermined subset size?
- How do the final test losses compare between CRAIG and the original algorithms?
- How do the relative distances (rather than the absolute distance) to the solution behave?
- How does CRAIG perform over multiple epochs? How does the algorithm transition when the subset is changed (as in neural networks)?
- Why does CovType appear more stable with the shuffled version over the other datasets? Is the stability related to the distribution of the weights $\gamma_j$?

Some grammatical errors/typos/formatting issues:

- Equation (9) needs more space between the $\forall x, i, j$ and the rest of the equation.
- What is $\Delta$ on page 5? Is it supposed to be $F$?
- On page 8, And should not be capitalized.
- Page 14, prove not proof
- Page 14, subtracting not subtracking
- Page 16, cycle not cycke

Overall, although I like the ideas in the paper, the paper still needs some significant amount of refining in terms of both writing and theory, as well as some additional experiments to be convincing. If the comments I made above were addressed, I would be open to changing my decision.

**Experience Assessment:**

I have read many papers in this area.

**Review Assessment: Checking Correctness Of Derivations And Theory:**

I assessed the sensibility of the derivations and theory.

**Review Assessment: Checking Correctness Of Experiments:**

I assessed the sensibility of the experiments.

**Review Assessment: Thoroughness In Paper Reading:**

I read the paper thoroughly.

---

> ### Author Response · Authors · 2019-11-15
> **Detailed Responses**
>
> We thank the reviewer for insightful feedback and for acknowledging the novelty and the algorithmic strength of our work. The reviewer asks great questions, and we provide detailed answers below.
>
> RE: Approximation of $d_{ij}$s
> Better estimation of $d_{ij}$s results in a smaller error in estimating the full gradient and a guarantee for converging to a closer neighborhood of the optimal solution. However, as shown in our experiments, in practice we do not need tight upper-bounds to get a satisfactory solution. For neural networks, we use the gradient of the loss with respect to the pre-activation outputs of the neural network as an upper bound on the per-sample gradients. As discussed in detail in Section 3.2 and Appendix B of (Katharopoulos & Fleuret ‘19), this upper-bound depends on the time step. Thus, CRAIG updates the subset at the beginning of every epoch. Our experimental results and the experiments of (Katharopoulos & Fleuret ‘19) confirm that this upper-bound is indeed useful in practice.
>
> RE: Choice of $\epsilon$
> In practice, we often have a limited time/budget that determines the size of the subset and hence the value of $\epsilon$. However, with larger errors in estimating $d_{ij}$s, we need to use a larger value for $\epsilon$, otherwise CRAIG may select a much larger subset.
>
> RE: Distribution of points obtained by CRAIG
> The gradient over the subset found by CRAIG is not an unbiased sample of the full gradient. The weighted subset provides an unbiased estimate of the full gradient only for $\epsilon=0$. At the same time, for clusterable datasets we may get $\epsilon=0$ for a subset that is much smaller than the full dataset.
>
> RE: Generalization performance
> While we do not have a formal proof in the paper, intuitively the subset acts as regularizer and provides a smoother decision boundary compared to the full dataset. As a result we expect to get a better generalization performance. Our experimental results confirm the superior generalization performance of CRAIG compare to the baselines (c.f. Fig. 1, 4, 6 in the Appendix C).
>
> RE: Subset $S$ in page 4
> $S$ is fixed in both sides of Eq 6. In fact, the subset $S$ that minimizes the right hand side of Eq. 6 is upper-bounding the minimum error in the left hand side of Eq. 6.
>
> RE: $k$ in Theorem 1, 2
> We thank the reviewer for pointing this out. The limit of $k \rightarrow \infty$ only holds for the second case where $0< \tau <1$. The other two cases where $\tau = 1$ or $\tau = 0$, show distance to the optimum after $k$ epochs (for arbitrary $k$). We modified the theorems accordingly.
>
> RE: Steplength
> The only requirement is $\alpha < \frac{1}{\mu}$.
>
> RE: Distance to the solution
> Since we have $\epsilon$ error in estimating the full gradient, every incremental movement could be off by $\alpha_k \epsilon$. Therefore, we are only guaranteed to converge to a neighborhood of the optimal solution. The size of this neighborhood depends on $\epsilon$ which decreases by increasing the size of the subset $S$. The result is the same if one starts from $||x_k-x_*||^2 > 2\epsilon R$.
>
> RE: Step length and regularization
> We repeated our largest convex experiments with carefully tuned stepsizes for SGD when the subset is found by 1) CRAIG, 2) uniformly at random with weights equal to 1, and 3) uniformly at random with all the weights equal to $|V|/|S|$. Figure 5 in Appendix C confirms the superior performance of CRAIG over random weighted and unweighted subsets. It can be observed that for carefully tuned stepsizes random subsets and random subsets upweighted by $|V|/|S|$ show similar performance. This confirms the quality of per-element stepsizes chosen by CRAIG. For IG we ran multiple trial and reported the average (we added error bars to Figure 5 in Appendix C, and will add to all the figures). For CRAIG, only one run is reported as there is a deterministic ordering on the elements of the subset. Our choice of regularizer is consistent with the SVRG paper.
>
> RE: Choice of $\epsilon$
> $\epsilon$ is the worst case estimation error of approximating the full gradient. In practice, this could be much larger than the actual error. Therefore, in our experiments we used a predetermined subset size.
>
> RE: CRAIG over multiple epochs
> For convex loss functions, CRAIG finds the subset as preprocessing and uses the same subset for all the epochs. For neural networks, the upper bound on the gradient norms depends on the time step t and changes as the model changes. Therefore, we use the upper bounds to get a different subset at the beginning of every epoch and only train on the subset. The transition is similar to the convex case, i.e., $x^{k+1}_0 = x^k_{|S|-1}$.
>
> RE: Test loss, relative distance and weight distribution
> We will add test loss and relative distance plots to the final version. This is a great observation by R1 that the weight distribution of Covtype is more uniform than the other 2 datasets. We will add this discussion to the final version.

---

### Official Review · AnonReviewer4 · 2019-11-03
**Official Blind Review #4**

**Rating:** 3

**Review:**

The paper proposes a theoretically founded method to generate subsets of a dataset, together with corresponding sample weights in a way that the average gradient of the subset is at most epsilon far from the average gradient of the full dataset. Given such a subset, the authors provide theoretical guarantees for convergence to an epsilon neighborhood of the optimum for strongly convex functions. The proposed algorithm to create such a subset is a greedy algorithm that relies on parameter independent similarities between samples, namely similarity scores that are true regardless of the current value of the function parameters.

Although I find the approach interesting, I have three main concerns with the proposed method.
1. The experimental setup is lacking significant information, baselines and baseline tuning (see below for more in depth comments).
2. The proposed upper bound which has been used for a similar purpose by [1] becomes nonsensical in high dimensions and although for [1] this would mean sampling with a non optimal sampling distribution for CRAIG it means converging very far from the optimum. What are the values of epsilon that you observe in practice?
3. I do not see how CRAIG would be applied to deep learning. The argument in section 3.4 is that the variance of the gradient norm is captured by the gradient of the last layer or last few layers, however this is true given the parameters of the neural network. The gradients can change arbitrarily after a very small number of parameter updates as shown by [2].

Experimental setup
----------------------------

For the case of the convex problems, the learning rate is not tuned independently for each method. Even more importantly the stepsizes of CRAIG are all numbers larger than 1 so the expected learning rate is multiplied by the average step size. This makes it difficult to understand whether the speedup is due to a larger learning rate or due to CRAIG. Similarly for figure 3 the result could be due to a non decreasing step size because of \gamma_j while for CRAIG \gamma_j are ordered in decreasing order.

In addition, there is no experimental analysis of the epsilon bound and the actual difference of the gradients for the subset and the full dataset. There are also no baselines that use a subset to train. A comparison with a baseline that uses 1. random subset or 2. a subset selected via importance sampling from [1] would contribute towards understanding the particular benefits of CRAIG.

Regarding the neural network experiments:
1. There is no explicit definition of the similarity function used for the case of neural networks. If we assume based on 3.4 that the algorithm requires an extra forward pass in the beginning of every epoch there should be visible steps in Figure 3 where time passes but the loss doesn't move.
2. 2000 seconds and 80% accuracy on MNIST points towards a mistake on the implementation of the training. On a laptop CPU it takes ~15s per epoch and achieves ~95% test accuracy from the first epoch for the neural network described.
3. Similarly 80% accuracy on CIFAR10 is sufficiently low for Resnet-56 to be alarming.

[1] Zhao, Peilin, and Tong Zhang. "Stochastic optimization with importance sampling for regularized loss minimization." international conference on machine learning. 2015.
[2] Defazio, Aaron, and Léon Bottou. "On the ineffectiveness of variance reduced optimization for deep learning." arXiv preprint arXiv:1812.04529 (2018).

**Experience Assessment:**

I have published one or two papers in this area.

**Review Assessment: Checking Correctness Of Derivations And Theory:**

I assessed the sensibility of the derivations and theory.

**Review Assessment: Checking Correctness Of Experiments:**

I carefully checked the experiments.

**Review Assessment: Thoroughness In Paper Reading:**

I read the paper thoroughly.

---

> ### Author Response · Authors · 2019-11-15
> **Detailed Responses**
>
> We thank the reviewer for acknowledging the technical aspects and the theoretically founded nature of the work. Based on reviewer’s valuable feedback we conducted a number of additional experiments, which further validate the efficacy of our CRAIG framework, and further strengthen the paper.
>
> RE: Learning rate and baseline
> We repeated our largest convex experiments with carefully tuned stepsizes for SGD when the subset is obtained by 1) CRAIG, 2) uniformly at random with weights equal to 1, and 3) uniformly at random with all the weights equal to $|V|/|S|$. Figure 5 in the Appendix shows the superior performance of CRAIG over random weighted and random unweighted subsets. It can be observed that for carefully tuned stepsizes random subsets and random subsets upweighted by $|V|/|S|$ show similar performance. This confirms that quality of per-element stepsizes chosen by CRAIG. Although we do not report the tuning time in this Figure, we note that tuning the initial rate and the decay factor of the SGD learning rate is significantly faster on the subset found by CRAIG compared to the full dataset. We also emphasize that the weight of ordered subsets produced by CRAIG is not necessarily a decreasing sequence. Indeed, some elements that comes earlier in the ordering has a smaller weight than the elements later in the ordering. We will clarify this in the manuscript and update the remaining experiments with carefully tuned step sizes.
>
> RE: Difference between gradients of the subset vs full data
> Calculating the actual difference between the gradient of the subset and the full dataset requires computing the full gradient for every $x \in \mathcal{X}$. Even estimating this difference by sampling a relatively small number of vectors $x \in \mathcal{X}$ is prohibitively expensive. On the other hand, by minimizing the upper-bound L(S) in Eq. 8, we are guaranteed to get a subset that closely estimates the full gradient for all $x \in \mathcal{X}$. Note that $\epsilon$ makes a trade off between cost and the quality of the obtained model. With a limited budget, CRAIG is guaranteed to provide us with a representative subset to train from big datasets.
>
> RE: Similarity function for NN and extra forward pass
> For neural networks, following (Katharopoulos & Fleuret ‘19), CRAIG uses the gradients of the loss with respect to the pre-activation outputs of our neural network as an upper bound on the per-sample gradients. The algorithm requires an extra forward pass in the beginning of every epoch to calculate the upper-bounds. Hence, the subset is updated for every epoch. The y-axis in Figure 3 shows loss/accuracy vs. time *after every epoch*. The time for a forward pass (~1 sec for MNIST and ~3sec for ResNet56) is negligible compared to the time of a backward pass, and would not be easily visible in the Figure. We will update the figures of MNIST and CIFAR10 to include this detail.
>
> RE: Accuracy and run time of MNIST and CIFAR10
> We thank the reviewer for the feedback. Using a more appropriate learning rate and bigger batches, we indeed got the expected result. Figure 6 in the Appendix C shows that SGD on the subset found by CRAIG achieves the same training loss as SGD on the full dataset. At the same time, we get a considerably better generalization performance. We will repeat our experiments on CIFAR10 with ResNet-56 using tuned learning rates to achieve optimal performance. We believe that based on our theory and as shown by the new MNIST experiment, CRAIG will obtain superior performance compared to the baselines.
>
> RE: Upper-bounding gradients in high dimensional space
> For higher dimensions, authors in “On the Surprising Behavior of Distance Metrics in High Dimensional Space“ suggest using a 1-norm (or even an alpha-fractional value for p in $||x||_p$, meaning $0<p<1$ when the dimensionality of x gets high). For this scenario, the 1-norm can be used instead of the 2-norm in Eq. 6 to find the near optimal subset. We note that the upper-bound on the gradient distance used by CRAIG has been used in several other works, including “Variance Reduced Stochastic Gradient Descent with Neighbors”, NIPS’15, and “Exploiting the Structure: Stochastic Gradient Methods Using Raw Clusters” NIPS’16, and has shown satisfactory performance even on high-dimensional datasets.
>
> RE: Upper-bounding gradients in neural networks
> As discussed in Section 3.4, this upper-bound is valid given the parameters of the neural network and hence for non-convex loss functions we update the subset at the beginning of every epoch. Our experiments and the experimental results of (Katharopoulos & Fleuret ‘19) confirm that this upper-bound is indeed useful in practice. For cases where the gradients may change too quickly, we can update the subset more than once during every epoch. As upper-bounds on the normed gradient distances can be obtained by a forward pass, this would still be considerably faster than backpropagation required for training on the full dataset.

---

### Decision · Program_Chairs · 2019-12-19

**Decision:**

Reject

**Comment:**

This paper investigates the practical and theoretical consequences of speeding up training using incremental gradient methods (such as stochastic descent) by calculating the gradients with respect to a specifically chosen sparse subset of data.

The reviewers were quite split on the paper.

On the one hand, there was a general excitement about the direction of the paper. The idea of speeding up gradient descent is of course hugely relevant to the current machine learning landscape. The approach was also considered novel, and the paper well-written.

However, the reviewers also pointed out multiple shortcomings. The experimental section was deemed to lack clarity and baselines.  The results on standard dataset were very different from expected, causing worry about the reliability, although this has partially been addressed in additional experiments. The applicability to deep learning and large dataset, as well as the significance of time saved by using this method, were other worries.

Unfortunately, I have to agree with the majority of the reviewers that the idea is fascinating, but that more work is required for acceptance to ICLR.